# Neural Correlates of Positive Outcome Expectancy for Aggression: Evidence from Voxel-Based Morphometry and Resting-State Functional Connectivity Analysis

**DOI:** 10.3390/brainsci14010043

**Published:** 2023-12-31

**Authors:** Jia-Ming Wei, Ling-Xiang Xia

**Affiliations:** 1Research Center of Psychology and Social Development, Faculty of Psychology, Southwest University, Chongqing 400715, China; wjmgallagher@foxmail.com; 2Key Laboratory of Cognition and Personality (Southwest University), Ministry of Education, Chongqing 400715, China

**Keywords:** positive outcome expectancy, aggression, posterior cingulate cortex, temporoparietal junction, medial prefrontal cortex, VBM, RSFC, prediction analysis

## Abstract

Positive outcome expectancy is a crucial cognitive factor influencing aggression, yet its neural basis remains unclear. Therefore, the present study combined voxel-based morphometry (VBM) with a resting-state functional connectivity (RSFC) analysis to investigate the brain correlates of positive outcome expectancy in aggression in young people. In the VBM analysis, multiple linear regression was conducted to explore the relationship between individual differences in aggressive positive outcome expectancy and regional gray matter volume (GMV) among 325 undergraduate students. For the RSFC analysis, seed regions were selected based on the results of the VBM analysis. Subsequently, multiple linear regression was employed to examine whether a significant correlation existed between individual differences in aggressive positive outcome expectancy and the RSFC of seed regions with other brain regions in 304 undergraduate students. The findings indicated that aggressive positive outcome expectancy was positively correlated with GMV in the posterior cingulate cortex (PCC), right temporoparietal junction (TPJ), and medial prefrontal cortex (MPFC). Moreover, it was also positively associated with RSFC between the PCC and the left dorsolateral prefrontal cortex (DLPFC). The prediction analysis indicated robust relationships between aggressive positive outcome expectancy and the GMV in the PCC, right TPJ, as well as the RSFC between the PCC and the left DLPFC. Our research provides the initial evidence for the neural basis of positive outcome expectancy in aggression, suggesting the potential role of the PCC as a hub in its neural network.

## 1. Introduction

Cognitive factors play a crucial role in aggression, influencing both the enactment of aggressive behavior and the formation of aggressive traits [1,2]. According to the Social Information Processing model, individuals undergo a series of cognitive processes before engaging in aggressive behavior. This series includes encoding and interpreting situational cues, clarifying goals, accessing or constructing responses, making response decisions, and behavioral enactment. Cognitive bias at any stage of these processes may potentially facilitate aggressive behavior [3]. Within studies on aggressive cognition, positive outcome expectancy (POE) is regarded as a pivotal cognitive bias influencing aggression, drawing sustained attention from researchers [4,5,6]. In previous research, the POE of aggression has been described as the belief that engaging in aggressive behavior can yield rewards or benefits for oneself, such as improved social standing and satisfying emotional experiences [7,8]. Therefore, in the present study, we characterize aggressive POE as the cognitive tendency to anticipate gaining desired outcomes through harming others [4,6].

In comparison to other aggressive cognitive biases, such as hostile attribution bias, aggressive POE exerts a broader influence on aggression. Aggression is commonly classified into reactive aggression and proactive aggression based on motivational differences. Reactive aggression involves retaliatory or defensive aggression triggered by provocation, whereas proactive aggression refers to instrumental aggression pursued for personal gain or goals in the absence of provocation [9,10]. Hostile attribution bias is believed to primarily influence reactive aggression [11], whereas POE may serve as a common motivator for both reactive and proactive aggression. The Social Learning Theory [12] and the Social Interaction Theory [13] propose that aggression is generally motivated by POE. Empirical evidence also indicates associations between aggressive POE and both reactive and proactive aggression [14,15]. Additionally, aggressive POE is linked to mental health problems, such as trauma symptoms, and poses obstacles to interventions addressing aggression [5,16]. Therefore, investigations into aggressive POE hold substantial theoretical significance and practical relevance. Despite significant progress in behavioral studies on aggressive POE, its neural mechanisms remain poorly understood. The brain plays a pivotal role as a neural basis for individual differences in mental processes [17,18,19]. Thus, the primary aim of this study was to explore the brain correlates of aggressive POE utilizing magnetic resonance imaging (MRI) methods.

Structural MRI is an effective tool for capturing individual differences in the neuroanatomical correlates of mental processes [19,20,21]. In neuroscience research, gray matter volume (GMV) serves as a typical index of brain structure, and previous studies have indicated that the individual differences in mental processes can be reflected in GMV [22,23]. Additionally, resting-state functional connectivity (RSFC) is also a useful neuroimaging index for depicting individual differences in mental processes. In contrast to structural connectivity methodologies (i.e., diffusion tensor imaging and DTI), which explore the orientation, microstructural intricacies, and connectivity of nerve fibers within the white matter [24], RSFC captures functionally synchronous changes in blood flow fluctuations among different brain regions [25,26]. This approach efficiently reveals potential neural connection features that underlie mental processes [27,28]. Thus, our intention was to investigate the brain correlates associated with individual differences in aggressive POE using GMV and RSFC as brain indexes. 

Despite the lack of direct neurobiological evidence for aggressive POE, several studies have indicated potential brain substrates related to its mental components and essential attributes [29,30,31], thereby inspiring assumptions about the key brain correlates associated with individual differences in aggressive POE. We posit that aggressive POE consists of two fundamental mental components: the cognitive association of “aggression-positive outcomes” and the anticipatory imagination of potential positive outcomes resulting from aggression. 

Firstly, outcome expectancy is comprised of cognitive associations between behavior and outcomes established through past experiences or observational learning [32,33], existing as a belief or schema stored in long-term memory [3,34]. In other words, associative learning and schematic memory are critical components of aggressive POE. Thus, we postulate that the brain regions related to associative learning and schematic memory may serve as neural correlates of aggressive POE. Secondly, outcome expectation, as a form of prospection, is future-oriented cognition, involving imagination and reasoning about the potential outcomes of enacting a specific behavior [35,36,37]. Imagining future events or speculating about potential behavioral outcomes relies on the retrieval of relevant information from memory [36,38]. For example, the social information processing model suggests that an individual’s learned outcome expectancy schema in long-term memory can guide the ongoing cognitive processing of aggression (e.g., predicting the potential outcomes of enacting an aggressive response) [2,3,34]. Therefore, the brain regions involved in self-centered autobiographical memory and future episodic thinking may be the brain correlates of aggressive POE, and these two mechanisms share a highly overlapping brain basis [31,36,38]. On the other hand, the anticipatory imagination of aggressive outcomes is predominantly concerned with personal outcomes (e.g., “will I obtain what I desire?” and “how will I feel?”) and social outcomes (e.g., “how will others evaluate me?” and “will others like me?”) [7,34,39]. Hence, the brain correlates of aggressive POE may also include the brain regions related to reward anticipation and inferring one’s own and others’ mental states, i.e., mentalizing or the theory of mind [30,40].

Based on the aforementioned arguments, we hypothesized that the ventromedial striatum (VS) and certain regions in the default mode network (DMN) are potential brain correlates of aggressive POE, as these regions have been implicated in associative learning, schema memory, and future imagination [29,30,41,42,43,44]. Firstly, the VS underpins the neural basis of reward-based associative learning [43,45]. Additionally, the hippocampus (HC) is a key brain region for paired-associate learning [46,47] and schema memory [42,48,49]. Therefore, the VS and hippocampus are likely to be implicated in the formation of “aggression-positive outcome” associations. Secondly, the DMN comprises two subsystems, along with the midline core hubs of the posterior cingulate cortex (PCC) and the anterior medial prefrontal cortex (aMPFC). The medial temporal lobe (MTL) subsystem, which includes the hippocampus/parahippocampal cortex, retrosplenial cortex, ventral MPFC, and posterior inferior parietal lobule, is primarily responsible for memory retrieval, future imagination, and episodic simulation. On the other hand, the dorsal medial prefrontal cortex (dMPFC) subsystem, including the dMPFC, temporoparietal junction, lateral temporal cortex, and temporal pole, mainly relates to self-related social cognition, including mentalizing and social reasoning [29,30,31,40]. The midline core hubs share functional properties of both subsystems [29,30]. Moreover, the core hubs of the DMN are involved in the valuation and emotional representation of anticipated events, indicating the motivational salience and personal significance of these events [29,36]. For example, during reward anticipation [50,51] and imagining goal achievement [52], activation of the PCC and aMPFC is significantly increased. Taken together, the DMN may be the brain correlate for imagining or inferring the potential positive outcomes of aggressive responses based on prior “aggression-positive outcome” schema.

On the other hand, as POE is an important driver of aggression [3,11], its brain correlates may involve regions related to approach motivation. The neuroscience literature shows that the left dorsolateral prefrontal cortex (DLPFC) is closely associated with approach motivation [53]. A recent study demonstrated that applying anodal transcranial direct current stimulation (tDCS) to the left DLPFC enhances participants’ approach motivation, leading them to allocate more effort towards rewards [54]. Brain indexes of the left DLPFC (e.g., gray matter density and BOLD signals) also exhibit a stable correlation with aggressive tendencies [55,56] and factors that facilitate aggression (e.g., Machiavellianism) [20,21]. Moreover, the VS could potentially serve as a neural correlate of aggressive POE. As part of the dopamine neural circuitry, the VS is associated with encoding the value of expected rewards, thereby providing motivational incentives to individuals [57,58,59,60]. This implies its potential role as the neural basis for the motivational effect of aggressive POE.

Considering the primary mental components and the motivational role of aggressive POE, we hypothesized that its brain correlates involve the VS, DMN regions (including the PCC, aMPFC, dMPFC, and hippocampus), and left DLPFC. To test this hypothesis, we first examined the correlations between the GMV of these regions and aggressive POE using voxel-based morphometry (VBM) [61] at the whole-brain level. Furthermore, previous studies have shown that individual differences in mental processes, as observed in the structure of different brain regions, are also reflected in the RSFCs of these regions with other brain regions [55,62]. Thus, we further explored the correlations between aggressive POE and the RSFCs of the clusters found in the VBM analysis.

## 2. Materials and Methods

### 2.1. Participants

A total of 325 (110 males and 215 females) healthy, right-handed Chinese college students (aged 18–33 years old, *M*_age_ = 20.58 years, *SD*_age_ = 1.56 years) were recruited for this study. All participants gave informed consent, and none of them had a history of neurological or psychiatric disorders. All of the 325 participants were included in the VBM analysis. However, during the RSFC analysis, 21 participants were removed from the study due to excessive head motion (translational parameters > 3 mm or rotational parameters > 3 rad). Thus, 304 participants (107 males and 197 females, *M*_age_ = 20.69, *SD*_age_ = 1.58) met the inclusion criteria and were included in the final RSFC analysis. The study was approved by the Research Project Ethical Review Committee of the Faculty of Psychology in Southwest University. All the participants received monetary compensation.

### 2.2. Assessment of Aggressive Positive Outcome Expectancy

Aggressive POE was assessed using the Social Emotional Information Processing questionnaire (SEIPQ) [39]. The SEIPQ included 8 hypothetical social situations designed to elicit aggressive behavior, such as a friend disclosing your private information or a colleague accidentally spilling coffee on you. Participants were instructed to imagine themselves in these scenarios and generate aggressive responses. Following this, they were asked to respond to a series of questions related to aggressive cognition, such as outcome expectancy for aggression. The SEIPQ comprised a set of 64 items that measured aggressive outcome expectancy (e.g., “if you acted this way, how likely is it that you will get what you want?, how likely is it that others will respect you?, how would you feel about yourself? and how likely is it that others will like you?”), each utilizing a 4-point Likert-type response format ranging from 1 to 4. Higher scores indicated that participants believe that aggression can lead to positive outcomes for themselves, such as gaining respect. The aggressive outcome expectancy items demonstrated good internal consistency in the present study, with a Cronbach’s alpha coefficient of 0.94 and 0.93 in the VBM and RSFC analysis, respectively.

### 2.3. Imaging Data Acquisition 

All structural and resting-state MRI images were acquired using a Siemens 3T scanner (Siemens Magnetom Trio TIM, Erlangen, Germany). 

High-resolution T1-weighted anatomical images were acquired using a Magnetization Prepared Rapid Acquisition Gradient Echo (MPRAGE) sequence (TR = 1900 ms, TE = 2.52 ms, inversion time = 900 ms, flip angle = 9 degrees, thickness = 1 mm, number of slices = 176, resolution matrix = 256 × 256 mm^2^, and voxel size = 1 × 1 × 1 mm^3^). The brain structure scan lasted for 4.5 min. 

Resting-state functional images were acquired using T2-weighted gradient-echo echo planar imaging (EPI) sequences (TR = 2000 ms, TE = 30 ms, resolution matrix = 64 × 64, flip angle = 90 degree, FOV = 220 × 220 mm^2^, slice gap = 1.0 mm, slice thickness = 3 mm, voxel size = 3.4 × 3.4 × 4 mm^3^, and number of slices = 32). Before the scan, participants were not involved in any experimental task or questionnaire survey and we stressed the importance of not engaging in deliberate thinking during the scan. During the scanning session, the participants were instructed to close their eyes and relax while remaining awake. This process lasted for 8 min and consisted of 240 volumes.

After the MRI data was acquired, the participants were instructed to complete the SEIPQ online within a week, after which they were free to leave the laboratory.

### 2.4. VBM Analysis

#### 2.4.1. Structural MRI Data Pre-Processing 

All pre-processing steps for the structural data were conducted following Ashburner’s guidelines (2007) [63]. The pre-processing of structural MRI data was conducted utilizing SPM12 (Wellcome Department of Cognitive Neurology, London, UK; www.fl.ion.ucl.ac.uk/spm/, accessed on 6 December 2019) in MATLAB 2016a (Mathworks Inc., Natick, MA, USA). 

First, the quality of the images was assessed by examining any visible artifacts or gross anatomical abnormalities. Second, the structural images were manually co-registered and reoriented to align with the anterior commissure–posterior commissure line. Third, all the images were segmented into gray matter (GM), white matter (WM), and cerebrospinal fluid (CSF) using the New Segmentation method. Fourth, the gray matter images underwent registration, normalization, and modulation using the DARTEL (Diffeomorphic Anatomical Registration Trough Exponential Lie algebra) tool. DARTEL registration involves computing the specific template using the average tissue probability maps obtained from all the participants, warping each participant’s segmented maps into the specific template. This procedure was repeated until a best study-specific template was generated. The image intensity of each voxel was modulated using the Jacobian determinants to conserve the regional differences in the absolute amounts of GMV. Later, the registered gray matter images were transformed into Montreal Neurological Institute (MNI) space. Finally, the images were smoothed with an 8 mm full width at half maximum (FWHM) Gaussian kernel to increase the signal-to-noise ratio. 

#### 2.4.2. Structural MRI Data Statistical Analysis

Statistical analyses for the GMV data was performed using SPM12 software (v6909). In the whole-brain analysis, multiple linear regression was performed between GMV and the scores of aggressive POE in the sample including all participants (n = 325), with gender and age as nuisance covariates. Multiple comparison correction was performed to test the significance of the correlation between GMV and aggressive POE. Specifically, the voxel-level significance threshold was set at *p* < 0.001 and the cluster-level significance threshold was set at *p* < 0.05 using the family-wise error (FWE) method. The surviving regions were saved as regions of interest (ROIs), from which the mean GMV was extracted using the REX Toolbox (http://web.mit.edu/swg/software.htm) (accessed on 12 October 2018) for further prediction analyses.

### 2.5. RSFC Analysis

#### 2.5.1. Resting-State MRI Data Pre-Processing

Data pre-processing was conducted using Data Processing Assistant for Resting-State fMRI (DPARSF) (V5.3_220101) [64] software in the MATLAB platform. First, the first 10 volumes from each participant were removed to ensure fMRI signal stabilization. The remaining 230 volumes were corrected for temporal shifts between slices and head motion artefacts. Second, each image was spatially normalized to the standard MNI template with a resampled voxel size of 3 × 3 × 3 mm^3^. Third, to reduce the potential effect of physiological artefacts and head motion, white matter, cerebrospinal fluid signals, and Friston 24 parameters for head motion [65] were regressed out. Fourth, all the images were smoothed using an isotropic 8 mm FWHM Gaussian kernel. Fifth, the linear trend was removed to reduce physiological noise and a temporal band pass filter (0.01–0.1 Hz) was applied to reduce the effect of low-frequency drift and high-frequency noise [25]. Finally, 21 participants were excluded from the second level analysis due to excessive head motion (translational parameter > 3 mm or rotational parameter > 3 rad). The mean framewise displacement (FD) values of all participants were under 0.3 mm.

#### 2.5.2. Resting-State Functional Connectivity Data Statistical Analysis

First, based on the results of the VBM analysis, three seed regions (PCC, MNI coordinate: 9, −48, 23; right TPJ, MNI coordinate: 63, −47, 21; MPFC, MNI coordinate: 0, 41, 39) were defined for the RSFC analysis. Following previous studies [28,55], spherical seed regions with a radius of 6 mm were created and centered on these coordinates. Secondly, in the individual first-level analysis, Pearson’s correlation coefficients were calculated between the average blood-oxygen-level-dependent (BOLD) signal time course inside the seed regions and all the remaining voxels in the whole brain. These individual-level correlation coefficient maps were transformed into normally distributed z-value maps using Fisher’s r-to-z transformation. Thirdly, at the group-level, multiple regression analysis was conducted to identify the functional connectivities of the seed regions that were significantly correlated with aggressive POE. The age and gender of the participants were regressed out in the analysis as nuisance covariates. Multiple comparison correction was performed based on two-tailed Gaussian random field (GRF) theory [66] within a default whole-brain mask (voxel-level *p* < 0.001, cluster-level *p* < 0.05). For each participant, the mean correlation coefficients of functional connectivities that met the threshold of multiple comparison correction were extracted and utilized for the subsequent prediction analysis.

### 2.6. Prediction Analysis 

Following the previous studies [20,28], to confirm the robustness of the relationship between the brain index (mean GMV and RSFC values) and aggressive POE, a machine-learning approach, based on balanced cross-validation with linear regression, was employed. In the regression analyses, brain index was treated as the independent variable, whereas aggressive POE was treated as the dependent variable. Firstly, the data were divided randomly and equally into four folds, ensuring balanced distributions of the variables across the folds. Next, three folds were employed to build a linear regression model and the model was used in the fourth fold. Specifically, in the fourth fold, the independent variable (the GMV or RSFC value) was input into the model to obtain the predicted dependent variable (aggressive POE). The training process was repeated four times. Hence, the mean correlation coefficient between the observed and predicted values of the dependent variable (*r*(_predicted, observed_)) was computed. *r*(_predicted, observed_) indicates the extent to which the independent variable predicts the dependent variable.

A permutation test was employed to test the null hypothesis that there was no significant correlation between the POE of aggression and GMV or RSFC. First, 1000 surrogate datasets (D*_i_*) were generated by randomly pairing the independent and dependent variable data in the sample. Using the four-fold balanced cross-validation procedure, a *r*(_predicted, observed_)*_i_* coefficient was computed for each of the surrogate datasets. To determine the statistical significance (*p*-value) of the correlation between the independent and dependent variables, we counted the number of *r*(_predicted, observed_)*_i_* greater than *r*(_predicted, observed_) and then divided the count by 1000 (the number of D*_i_* datasets). If *p* < 0.05, the null hypothesis was rejected, indicating that the correlation between aggressive POE and GMV or RSFC was stable.

## 3. Results

### 3.1. VBM Results

To investigate the anatomical brain correlates of individual differences in aggressive POE, a multiple regression analysis was conducted, with age and gender entered as covariates. The results showed that aggressive POE was positively correlated with the GMV in the PCC (MNI coordinate: 9, −48, 23; Brodmann area 23, 31) and the right TPJ (MNI coordinate: 63, −47, 21; Brodmann area 40; see Figure 1, Table 1). In addition, when the threshold of multiple comparison correction was less strict (voxel-level *p* < 0.005), aggressive POE was positively correlated with the GMV of the MPFC (MNI coordinate: 0, 41, 39; Brodmann area 9; the cluster was extended to include dMPFC and aMPFC; see Figure 2, Table 1). However, inconsistent with our hypothesis, there was no significant correlation between aggressive POE and the GMV of the VS and hippocampus. Next, the GMV values of the PCC, right TPJ, and MPFC were extracted and correlated with aggressive POE scores. Aggressive POE exhibited significant positive correlations with the GMV values of the PCC (*r* = 0.28, *p* < 0.001), right TPJ (*r* = 0.26, *p* < 0.001), and MPFC (*r* = 0.27, *p* < 0.01; see Figure 1 and Figure 2).

A prediction analysis was performed to examine the robustness of the relation between regional GMV values and aggressive POE. The four-fold cross-validation analysis and permutation test indicated significant correlations between the GMV values of the PCC [*r*(_predicted, observed_) = 0.27, *p* < 0.001], right TPJ [*r*(_predicted, observed_) = 0.23, *p* < 0.001], and MPFC [*r*(_predicted, observed_) = 0.17, *p* < 0.01] with aggressive POE scores, suggesting that the associations between the GMV in the PCC, right TPJ, and MPFC and aggressive POE are reliable.

### 3.2. RSFC Results

To investigate the relationship between aggressive POE and the functional connectivities between seed regions identified in the VBM analysis and other brain areas, the RSFC behavior correlation analysis was conducted. The analysis indicated that only the RSFC strength between the PCC (seed region) and the left DLPFC (MNI coordinate: −21, 54, 24; Brodmann area 10; cluster sizes = 198, *t* = 5.15, *p* < 0.05, corrected; see Figure 3) was positively correlated with aggressive POE scores. Then, we extracted the mean RSFC value between the PCC and left DLPFC for each of the participants. The correlation analysis revealed a significant relationship between RSFC value and aggressive POE (*r* = 0.29, *p* < 0.001).

The prediction analysis showed that the mean RSFC value between the PCC and left DLPFC significantly correlated with aggressive POE scores [*r*(_predicted, observed_) = 0.25, *p* < 0.001]. This result suggests that the association between the RSFC of the PCC and left DLPFC and aggressive POE is robust.

## 4. Discussion

The aim of our study was to explore the neurobiological correlates of individual differences in positive outcome expectancy for aggression by combining VBM and RSFC analyses. The whole-brain analysis revealed that aggressive POE was positively associated with the GMV in the PCC, right TPJ, and MPFC, which included both the dMPFC and aMPFC. Moreover, the strength of functional connectivity between the PCC and left DLPFC was positively correlated with aggressive POE.

Consistent with our hypothesis, the brain correlates of individual differences in aggressive POE involve regions within the DMN. Firstly, the GMVs of the PCC and aMPFC were found to be correlated with aggressive POE scores. Aggressive POE is a future-oriented and self-related cognitive variable that is based on past experience. PCC and aMPFC, as core hubs of the DMN, are involved in both remembering past and imagining future personal events [30,38]. Both regions are also involved in reward anticipation [50,51], representing the subjective value and personal significance of anticipated events [29,36,67], which may underlie the brain basis of the motivational function of aggressive POE. On the other hand, the PCC may be implicated in the formation of beliefs about aggressive POE. The PCC is a vital node of the action–outcome learning system, which can simultaneously receive action-related information and value-related outcome information from other brain areas [68]. Aggressive POE is shaped by personal experiences of aggressive behavior and reinforcing consequences, such as dominance [12,16,34]. In other words, individual differences in learning the cognitive associations of “aggression-positive outcomes” may be related to the GMV of the PCC. In summary, the PCC and aMPFC are important neural correlates of individual differences in aggressive POE, with the PCC potentially playing a central role in its brain basis. 

Secondly, the whole-brain VBM analysis indicated that the GMV of the TPJ and dMPFC was positively correlated with aggressive POE. The TPJ and dMPFC are nodes in the DMN’s dMPFC subsystem, which are involved in functions such as mentalizing and social reasoning [29,40,69]. In most cases, aggressive behavior is goal-directed and socially disapproved [12,70]. Hence, aggressive POE involves the inference of one’s own mental state after engaging aggression, as well as social evaluation from others [34,39]. Our results indicated that the TPJ and dMPFC, which are associated with mentalizing, may constitute the structural brain basis underlying individual differences in aggressive POE. 

Thirdly, the RSFC analysis found that aggressive POE is positively related to the strength of functional connectivity between the PCC and the left DLPFC. The left DLPFC is associated with approach motivation [53,71], as demonstrated in a recent study that showed significant activation in this region when individuals engaged in aggressive responses under reward incentives [56]. Moreover, the PCC can receive “action–outcome” associations through back projections from the memory system [68]. In other words, regarding the results of the RSFC analysis, one possible explanation is that the high-strength PCC and left DLPFC RSFC values suggest that the PCC may transmit more signals representing the positive outcomes of aggression to the DLPFC. This may subsequently strengthen an individual’s motivation or inclination towards aggression. Thus, the functional connectivity between the PCC and the left DLPFC may underlie the brain basis for the motivational function of aggressive POE.

Finally, the present study did not identify any significant associations between aggressive POE and the VS or hippocampus. Previous studies have indicated that the VS and hippocampus are involved in associative learning or schema memory in functional MRI tasks [43,44,49]. However, in this study, outcome expectancy reflects a stable cognitive tendency rather than the process of combining stimuli or behavior with outcomes. On the other hand, the motivational role of the VS is closely linked to dopamine transmission. Yet, because this study did not employ a task-based fMRI design, capturing the dynamic processes of dopamine activity presents a challenge. Therefore, due to the limitations of brain indexes and psychological measures in our study, it is possible that the correlation between aggressive POE and these two regions could not be effectively demonstrated. 

In summary, our research indicates that the PCC, right TPJ, MPFC, and left DLPFC are reliably associated with individual differences in aggressive POE. Several studies have shown that these brain regions may related to an increased likelihood of aggression. For example, the PCC, right TPJ, MPFC, and left DLPFC are significantly activated in aggressive responses, and their morphological features can also predict aggressive tendencies [20,55,56,72,73]. Moreover, brain structural characteristics in these regions positively correlate with some negative traits that promote aggression, such as Machiavellianism [20,21,74]. These findings imply that the PCC, right TPJ, MPFC, and left DLPFC represent important components of the neural basis for the approach system of aggression. Therefore, enhanced aggressive POE, as a driving risk factor for aggression, may be associated with alterations in these brain regions. However, due to the cross-sectional design of this study, it remains uncertain whether aggressive POE shapes the neurobiological features of these brain regions or whether these brain regions contribute to the development of aggressive POE. Thus, we suggest that future research should utilize longitudinal designs and experimental paradigms to investigate the potential reciprocal causality relationship between aggressive POE and its brain correlates.

## 5. Limitations and Future Directions

Although our hypotheses were largely supported by the findings, some limitations of this study should be acknowledged. Firstly, due to the utilization of SEIPQ in this study, our findings predominantly reflect the neural correlates of explicit aggressive POE. The neural basis of implicit, automated POE in aggression requires further exploration. Secondly, despite the association between aggressive POE and reward processing, some crucial brain regions in the reward neural system, such as the VS and dorsal diencephalic conduction system (DDCS), did not exhibit significant correlations with aggressive POE in this study. The biological basis through which these brain regions influence reward processing involves the transmission of neurotransmitters, such as dopamine and gamma-aminobutyric acid (GABA). In other words, using GMV and RSFC as brain markers in this study might not reveal the potential correlation between the reward neural system and aggressive POE. Thus, we suggest that future studies should develop specific aggressive POE tasks and employ fMRI methodology to investigate the neural correlates of aggressive POE. Thirdly, the participants in this study were all healthy, well-educated college students. Investigating the neural correlates of aggressive POE in other specific samples, such as violent offenders and those addicted to aggression, may lead to the discovery of new pathological biomarkers.

## 6. Conclusions

The present study, for the first time, revealed neurobiological markers associated with aggressive POE by examining brain morphology and resting-state functional connectivity. Our findings were derived from a large-sample MRI dataset, underwent stringent multiple comparison corrections, and can be repeatedly verified using a machine learning approach. Therefore, our findings are robust. Our research indicated that the GMV in the PCC, right TPJ, and MPFC (the cluster extended to both dMPFC and aMPFC), and the strength of PCC and left DLPFC RSFC were positively correlated with individual differences in aggressive POE. The prediction analysis further confirmed the reliability of the above results. In summary, our study provides novel evidence for the neural basis of aggressive POE, and these findings highlight the importance of investigating the relationship between the default mode network and aggressive POE in future research. This research advances aggressive cognitive theory from a neurobiological perspective and can serve as a reference for neuro-based cognitive interventions in aggression.

## Figures and Tables

**Figure 1 brainsci-14-00043-f001:**
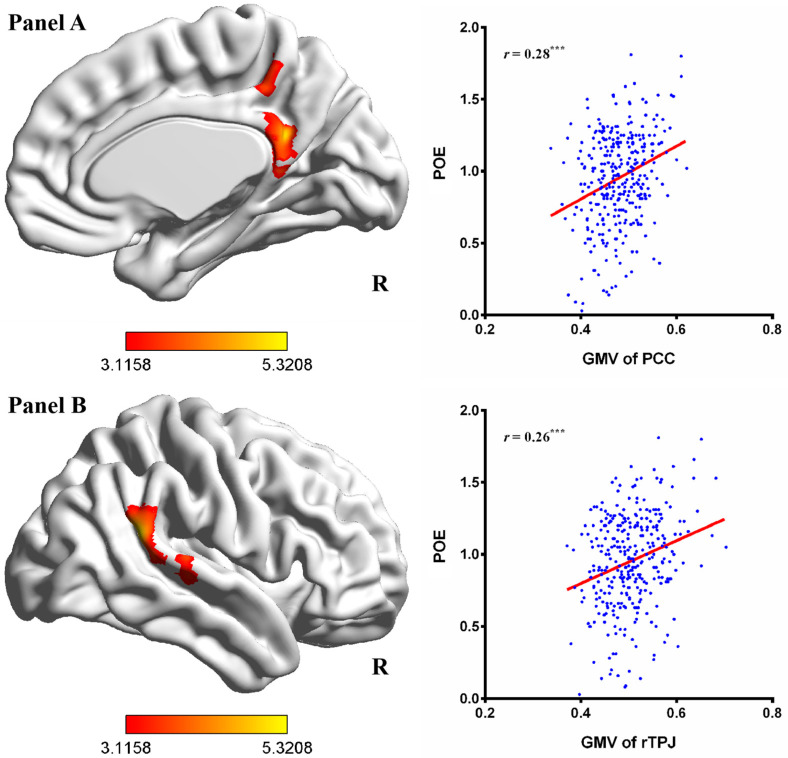
Correlations between regional GMV and aggressive POE. POE, positive outcome expectancy; PCC, posterior cingulate cortex; rTPJ, right temporoparietal junction. (**Panel A**) shows the correlation between aggressive POE and the GMV of the PCC; (**Panel B**) shows the correlation between aggressive POE and the GMV of the MPFC. The results were corrected using the FWE method (voxel-level *p* < 0.001, cluster-level *p* < 0.05). Note. ^***^
*p* < 0.001.

**Figure 2 brainsci-14-00043-f002:**
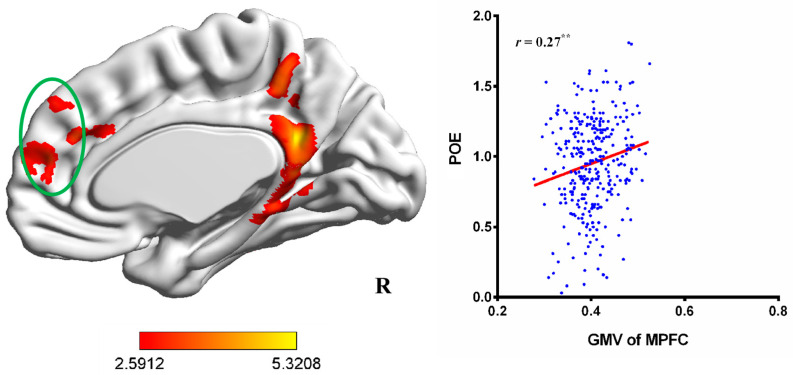
Correlations between the GMV of the MPFC and aggressive POE (the significant areas in the MPFC were presented in green circles, involving both dMPFC and aMPFC.). POE, positive outcome expectancy; MPFC, medial prefrontal cortex. The result was corrected using the FWE method (voxel-level *p* < 0.005, cluster-level *p* < 0.05). Note. ^**^
*p* < 0.01.

**Figure 3 brainsci-14-00043-f003:**
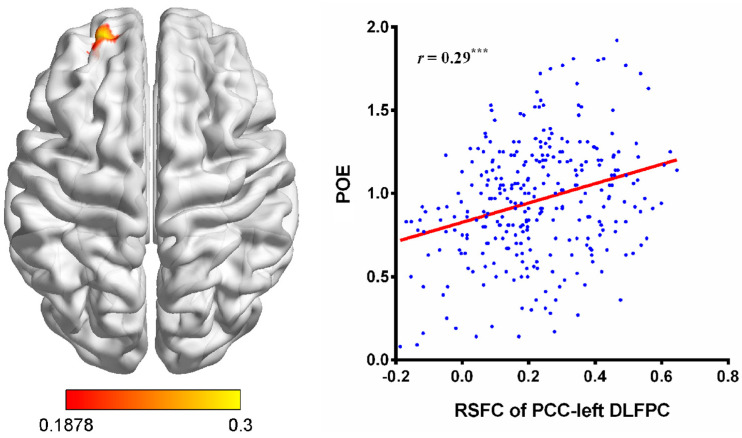
Correlations between the RSFC of the PCC and left DLPFC and aggressive POE. POE, positive outcome expectancy; PCC, posterior cingulate cortex; DLPFC, dorsolateral prefrontal cortex. Voxel−level *p* < 0.001, cluster-level *p* < 0.05, cluster size > 146, two−tailed GRF corrected. Note. ^***^
*p* < 0.001.

**Table 1 brainsci-14-00043-t001:** Regions where GMV values significantly associated with *aggressive* POE in a whole-brain analysis.

Regions	Brodmann Areas	Peak MNI Coordinates			Peak t-Value	Cluster Size
		*x*	*y*	*z*		
Only+						
PCC	BA 23, 31	9	−48	23	5.10	1889
Right TPJ	BA 40	63	−47	21	5.32	2223
MPFC ^a^	BA 9, 10 ^b^	0	41	39	3.821	2134

Note: PCC, posterior cingulate cortex; TPJ, temporoparietal junction; MPFC, medial prefrontal cortex; MNI, Montreal Neurological Institute; BA, Brodmann areas; Only+, the GMV of the regions positively correlated with aggressive POE. ^a^, the correlation between aggressive POE and the GMV of the MPFC was significant only when the voxel-level threshold was set at *p* < 0.005; ^b^, situated in the medial aspect of BA9 and BA10.

## Data Availability

Upon reasonable request, the research data can be obtained by contacting the corresponding author. Due to the expensive cost of collecting research data and the involvement of participants’ personal information, the data cannot be made publicly available.

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
