# Peer review of "Neural Correlates of Positive Outcome Expectancy for Aggression: Evidence from Voxel-Based Morphometry and Resting-State Functional Connectivity Analysis"

_brainsci, 2023, doi:10.3390/brainsci14010043_

Round 1
Reviewer 1 Report
Comments and Suggestions for Authors
The authors describe the relationship of a possible association between aggressive positive outcome expectancy (APOE) - measured with the Social Emotional Information Processing questionnaire (SEIPQ) - and structural differences as inferred from gray matter volume (GMV) and resting-state functional connectivity (RSFC) - measured with 3T MRI - in eventual 304 participants (107 males, 197 females), all healthy, right-handed Chinese college students (18-33 years old).
1 This can be improved by discussing many details of the method used in supplementary material that can be referred to in the main article.
2. Second, the designations of cortical areas are far too imprecise; insistence should be made on mentioning specific parts of Brodmann areas (BA).
3. This is especially true for the medial and dorsolateral prefrontal cortex. Reference can be made to the plea of Van Heukelum et al. (2020, doi: 10.1016/j.tins.2020.03.007).
4. Referent also wonders, why was connectivity within the dorsal diencephalic connective system (DDCS) not examined. This is definitely interesting within it examined the neuronal regulatory system for APOE. Both connectivity within the medial to lateral subdivisions could be looked at. If this is not feasible within the experimental setup, this should be mentioned among the limitations.
5. The authors describe in the discussion that no abnormality was found in the ventromedial striatum, but this is not discussed among the results.
6. Incidentally, no subcortical structure is mentioned which may be related to the equipment and experimental setup used. This problem should also be clearly discussed.
7. Finally, in the referent's opinion, it should be examined when the SEIPQ was taken in relation to the timing of the MRI scans and how it was avoided that the subjects' thoughts were not determined by the subject of the study. That they had to relax during the scan does not seem to referent sufficient in this regard.
Comments on the Quality of English LanguageReviewer does not understand the sentence "While the imagination of future events or episodic events is based on the retrieval of memory information [31,33]."(line 71-72).
Reviewer 2 Report
Comments and Suggestions for Authors
Dear Authors,
I read with interest your manuscript entitled “Neural Correlates of Positive Outcome Expectancy for Aggression: Evidence from Voxel-Based Morphometry and Resting-State Functional Connectivity Analysis”. The paper is very interesting, and I appreciate in particular the style and the method used for assertions, that are empirical and driven by evidence of correlations and probabilistic approach rather than statements oriented toward a semplistic and deterministic view of psychological dimensions and lived experiences. Specifically, the method of verbal reports or questionnaires represents a limited way to actual investigate some psychological dimensions, especially behaviours or attitude, without an ecological context. Only probabilistic and hypothetical assumptions can be formulated, due to indirect associations of variables operationalized and detected by interviews, with actual behaviours.
I report some comments below.
1) The term “APOE” may sound misleading as acronim, since it can refer to other concepts (i.e., Apoliprotein APOE). Please modify with another acronim.
2) I suggest also to rethink the term “mental variables” (see Introduction, line 52-53; line 58). It might be more appropriate to name as “mental processes”. Also, the following sentence should be better explained or changed: “Thus, APOE may be linked to brain regions involved in associative learning and schematic memory”. Maybe author would intend prospective memory, planning, or only a kind of memory related to automatic, implicit learned associations?
3) The methodological choice to investigate the brain correlates “in young people” should be specified in the abstract (i.e., line 13-14).
4) The topic of positive outcome expectancy is very meaningful as possible explanation of some aggressive behaviors. I suggest to introduce a brief perspective of social psychology, in particular in relation to some constructs as the hostile attribution bias and the fundamental attribution error. These hypothetical explanations deal with cognitive biases and have been put in relation with some aggressive behaviours as bullying. I believe a brief focus on this topic could be meaningful.
5) Similarly, Authors correctly mention the ventromedial striatum as an important brain region involved in motivation. To this purpose, I suggest to better specify the role of mesolimbic system in motivation. Did Authors analyze the hypothalamus or the nucleus Accumbens? Any finding raised about structural or functional implications for these regions? Can some constructs be implicated in the processes described by Authors, as intrinsic motivation and the self-determination theory? I suggest to consider a paper:
Di Domenico SI, Ryan RM. The Emerging Neuroscience of Intrinsic Motivation: A New Frontier in Self-Determination Research. Front Hum Neurosci. 2017 Mar 24;11:145. doi: 10.3389/fnhum.2017.00145.
6) Authors well argumented in limitations section that other paremeters, as cortical thickness, may be useful to further investigate their present findings. I suggest to briefly introduce a methodological distinction between the functional connectivity and structural connectivity (i.e., DTI) in the introduction. Any electrophsyiological studies with Event-Related Potentials (ERPs) have been carried out in cognitive psychology research for positive outcome expectancy? I wonder that some findings in this field may be significant.
7) I suggest to consider the following papers:
Neural correlates of aggression outcome expectation and their association with aggression: A voxel-based morphometry study Xinyu Gong, Bohua Hu, Liang Wang, Qinghua He, Ling-Xiang Xia. medRxiv 2023.08.26.23294598; doi: https://doi.org/10.1101/2023.08.26.23294598
Chester, D. S. (2017). The Role of Positive Affect in Aggression. Current Directions in Psychological Science, 26(4), 366-370. https://doi.org/10.1177/0963721417700457
Comments on the Quality of English Language
none
Round 2
Reviewer 1 Report
Comments and Suggestions for Authors
This reviewer is very grateful to the authors for their diligence in adapting the manuscript. Indeed, the authors misunderstood what I meant by transferring a significant piece of the description of methods to supplementary material. However, this is not a breaking point.
Many thanks for mentioning the Brodmann Area numbers. However, this brings to light an important shortcoming. Both BA 9 and BA 10 belong to the lateral prefrontal cortex (LPFC). BA 9 thus does not belong to the medial prefrontal cortex (MPFC) and thus the involvement of the MPFC is incorrectly mentioned in the abstract, results and discussion. The manuscript should be reviewed in its entirety by the authors to correct this throughout.
The above also raises an important question that will still need to be discussed. The MPFC (orbitofrontal, subcallosal and anterior cingular cortices), the hippocampal complex and the amygdaloid complex feed the ventral striatum and also provide important input to the DDCS. These structures are very important for generating and regulating the emotional response. Perhaps the authors can speculate why this is not reflected in their study. The resolution power is (apparently) very small. Does this play a role or is the absence related to other (theoretical) aspects of their experimental design?
